# Predictive Utility of Changes in Optic Nerve Sheath Diameter after Cardiac Arrest for Neurologic Outcomes

**DOI:** 10.3390/ijerph18126567

**Published:** 2021-06-18

**Authors:** Heekyung Lee, Joonkee Lee, Hyungoo Shin, Changsun Kim, Hyuk-Joong Choi, Bo-Seung Kang

**Affiliations:** 1Department of Emergency Medicine, Hanyang University Guri Hospital, 153, Gyeongchunro-ro, Guri, Gyeonggi-do 11923, Korea; massdt@naver.com (H.L.); xjkx@naver.com (J.L.); flyes98@naver.com (C.K.); airwaymanage@gmail.com (H.-J.C.); olivertw@hanyang.ac.kr (B.-S.K.); 2Department of Emergency Medicine, Graduate School of Medicine, Hanyang University, Seoul 04763, Korea; 3Department of Emergency Medicine, College of Medicine, Hanyang University, 222, Wangsimni-ro, Seongdong-gu, Seoul 04763, Korea

**Keywords:** heart arrest, optic nerve sheath diameter, patient outcome assessment

## Abstract

The optic nerve sheath diameter (ONSD) can help predict the neurologic outcomes of patients with post-cardiac arrest (CA) return of spontaneous circulation (ROSC). We aimed to investigate the effect of ONSD changes before and after CA on neurologic outcomes in patients with ROSC after CA using brain computed tomography (CT). The study included patients hospitalized after CA, who had undergone pre- and post-CA brain CT between January 2001 and September 2020. The patients were divided into good and poor neurologic outcome (GNO and PNO, respectively) groups based on their neurologic outcome at hospital discharge. We performed between-group comparisons of the amount and rate of ONSD changes in brain CT and calculated the area under the curve (AUC) to determine their predictive value for neurologic outcomes. Among the 96 enrolled patients, 25 had GNO. Compared with the GNO group, the PNO group showed a significantly higher amount (0.30 vs. 0.63 mm; *p* = 0.030) and rate (5.26 vs. 12.29%; *p* = 0.041) of change. The AUC for predicting PNO was 0.64 (95% confidence interval = 0.53–0.73; *p* = 0.04), and patients with a rate of ONSD change >27.2% had PNO with 100% specificity and positive predictive value. Hence, ONSD changes may predict neurologic outcomes in patients with post-CA ROSC.

## 1. Introduction

Ischemia/reperfusion cerebral injury after cardiac arrest (CA) may cause cerebral edema [1,2]. This results in an increase in intracranial pressure (ICP) and contributes to poor neurologic outcomes in patients with post-CA return of spontaneous circulation (ROSC) [3,4]. In these survivors, there is a need for the early detection of increased ICP and prediction of neurologic outcomes to facilitate appropriate post-resuscitation care [5]. This can help prioritize the allocation of limited medical resources to patients with expected good neurologic outcomes. There have been studies on various predictive factors for post-CA neurologic outcomes, including neurologic examination of brainstem reflexes, electrophysiological tests, and serum biomarkers, such as neuron-specific enolase and S-100B [6,7,8]. However, these have been recommended as prognostic factors at 72 h post-CA [8,9,10]. Moreover, early brain computed tomography (CT) of patients with post-CA ROSC may play a crucial role as a prognostic predictor. Further, the American Heart Association guidelines recommend early post-CA brain CT scans and confirm that a decrease in the gray-to-white matter ratio (GWR) can help predict neurologic outcomes [3,10,11,12]. Additionally, there have been studies regarding the role of the optic nerve sheath diameter (ONSD) on brain CT in predicting neurologic outcomes in post-CA survivors [13,14,15].

Previous studies have indicated the potential role of the ONSD on brain CT as a useful tool for non-invasive ICP measurement [16,17]. Additionally, recent studies have demonstrated that the ONSD on brain CT is useful for early neurologic outcome prediction through the evaluation of increased ICP in patients with post-CA ROSC [14,18]. Two recent meta-analyses confirmed the utility of the ONSD as a prognostic factor for neurologic outcomes in post-CA patients [19,20]. However, the methodology of these systematic reviews and meta-analyses is concerning [19,20]. The definition of poor neurologic outcome (PNO) included both Glasgow–Pittsburgh Cerebral Performance Categories (CPCs) 5 and CPC 3–5, and the outcome measurement time had a wide range (from hospital discharge to 6 months post-discharge). These limitations may have influenced the results of the studies. Furthermore, most of the studies included in the meta-analysis indicated that the sole use of the ONSD had limited predictive utility for prognosis. All these studies measured only the post-CA ONSD values without considering changes within individuals.

This study aimed to assess the differences between the pre- and post-CA ONSDs in patients with ROSC after CA using brain CT imaging. Additionally, we aimed to investigate the impact of the amount and rate of post-CA ONSD changes on the neurologic outcome at discharge.

## 2. Materials and Methods

### 2.1. Study Design and Population

This retrospective observational cohort study investigated brain CT scans of patients hospitalized after CA at a single university-affiliated hospital in Korea between January 2001 and September 2020. This study was approved by the Institutional Review Board of Hanyang University Guri Hospital (IRB No. GURI 2020-12-008). The requirement for informed consent was waived due to the retrospective nature of the study.

We included adult patients hospitalized after CA who underwent pre- and post-CA brain CT. The exclusion criteria were as follows: (1) transfer to another hospital after ROSC, (2) age <19 years, (3) traumatic/non-traumatic brain hemorrhage or brain tumor, (4) a history of ophthalmological disorders or surgeries that could affect the ONSD, and (5) the most recent pre-CA brain CT was performed at an age <19 years. Finally, eligible patients were divided into the good neurologic outcome (GNO) and poor neurologic outcome (PNO) groups based on their neurologic outcome at discharge; subsequently, we measured ONSD changes and performed between-group analysis. The primary outcome was the association between ONSD changes and the neurologic outcomes of patients hospitalized after CA.

### 2.2. Data Collection

We retrospectively collected the following data from electronic medical records: age; sex; comorbidities (hypertension, diabetes, and myocardial infarction); etiology (cardiac and respiratory); the location of the CA; whether the CA was witnessed; bystander cardiopulmonary resuscitation (CPR); the first monitored shockable rhythm; the CA duration, including the no-flow time (the time between CA and CPR initiation) and the low-flow time (the time between active CPR and ROSC); and administered targeted temperature management (TTM). Based on the medical records, we determined the interval between the latest pre-CA brain CT and ROSC (month), which was termed “CT to ROSC”, and between ROSC and post-CA brain CT (min), which was termed “ROSC to CT”. Additionally, we collected data regarding the neurologic outcomes at discharge using the Glasgow–Pittsburgh CPC. Based on the CPC scale, we defined GNO and PNO as a CPC of 1 or 2 and 3–5, respectively.

### 2.3. ONSD Measurements Using Brain CT

Brain CT scans were performed based on standard protocols using non-contrast 4 mm contiguous slices parallel to the orbital floor from the skull base to the vertex. The pre-CA and post-CA ONSDs were bilaterally measured at 3 mm behind the globe on brain CT using the picture archiving and communication system (PACS) ruler tool (PiView STAR, INFINITT, Seoul, Korea). Images were magnified at 450% and changed to the “mediastinum window (window width: 440; window level: 45) using the PACS tool. The ONSDs of the right and left eyes were averaged to obtain the mean value. All measurements were performed by emergency physicians blinded to the patient information, including the neurologic outcomes. Additionally, we calculated the amount and rate of ONSD change. We defined the amount of change as the difference between the pre-CA and post-CA ONSD. Moreover, the rate of ONSD change was calculated as follows:$$\mathrm{Rate}\;\mathrm{of}\;\mathrm{ONSD}\;\mathrm{change}=\left(\frac{\;\mathrm{Post}-\mathrm{CA}\;\mathrm{ONSD}\;-\;\mathrm{Pre}-\mathrm{CA}\;\mathrm{ONSD}}{\mathrm{Pre}-\mathrm{CA}\;\mathrm{ONSD}}\right)\times 100$$

We used the following CT equipment: SOMATOM Sensation 16, SOMATOM Definition DS, and SOMATOM Definition Edge (Siemens Healthcare, Erlangen, Germany). The following parameters were used: 120 kVp, 250–500 mAs, and 4 to 4.5 mm slice thickness. All CT images were stored in the Digital Imaging and Communication in Medicine format in the PACS.

### 2.4. Sample Size

We calculated the sample size based on a pilot study of 33 participants using G*Power (3.1.9.6; Heinrich Heine University, Düsseldorf, Germany). The mean ONSD of patients with GNO and PNO were 4.75 ± 1.45 mm and 5.63 ± 1.85 mm, respectively. The required sample size was calculated as 90 participants (effect size: 0.53; a-error: 0.05; power: 0.8); finally, considering a 10% drop-out rate, 99 participants were required.

### 2.5. Statistical Analysis

Continuous and categorical variables were reported as the median with interquartile range (IQR) and number with percentages, respectively. Normally distributed variables were analyzed using the Mann–Whitney U test and Wilcoxon rank sum test, while non-normally distributed variables were analyzed using the Shapiro–Wilk test. The chi-square test or Fisher’s exact test was used to analyze categorical variables. Statistical significance was set at *p* < 0.05. Multivariable analysis with logistic regression was used to determine the risk factors for poor neurologic outcomes, with adjustment for confounding variables found to be significant in univariate analysis. Variables with *p* < 0.2 in univariate analysis with the rate of ONSD change were included in the multivariable analysis. Further, the Hosmer–Lemeshow test was used to confirm the logistic model calibrations. The predictive performance of the main outcome was assessed using the area under the receiver operating characteristic curve ((ROC) AUC) with a sensitivity over 1 indicating- specificity. Results were obtained using the Youden Index and presented as a 95% confidence interval (CI) of AUC with sensitivity, specificity, positive predictive value (PPV), and negative predictive value (NPV). ROC analysis was performed using MedCalc Statistical Software (version 17.2, MedCalc Software, Ostend, Belgium), while the other statistical analyses were performed using SPSS software (version 25.0, IBM, Armonk, NY, USA).

## 3. Results

### 3.1. Baseline Characteristics

Among 145 post-CA survivors who underwent brain CT before and after CA, 49 patients were excluded as follows: 40 patients who were transferred to another hospital, seven patients with intracranial or subarachnoid hemorrhage, one patient with a brain tumor, and one patient aged ≤18 years. Finally, we enrolled 96 patients and allocated them to the GNO (*n* = 25, 26.0%) or PNO group (*n* = 71, 74.0%) (Figure 1).

Table 1 summarizes the patients’ demographics and clinical characteristics. The median age of the included patients was 70 (IQR: 58–79) years, and 56.3% were male. The GNO group was significantly younger than the PNO group. Moreover, the GNO group displayed a significantly higher frequency of cardiac etiology and shockable rhythm, as well as shorter no-flow and low-flow times, than the PNO group. Contrastingly, out-of-hospital cardiac arrest was more frequent among the PNO group.

### 3.2. Comparison of Pre-CA and Post-CA ONSDs

In both groups, the post-CA ONSD was significantly higher than the pre-CA ONSD (Figure 2). The pre-CA ONSD and post-CA ONSD were 5.06 and 5.50 mm, respectively, (*p* < 0.001) in the GNO group, and 5.07 and 5.72 mm, respectively, (*p* = 0.001) in the PNO group (Appendix A).

### 3.3. The Association between ONSD Changes and Neurologic Outcomes

Table 2 presents the between-group comparisons of the amount and rate of ONSD changes. There were no significant between-group differences in the pre-CA ONSD (5.06 vs. 5.07 mm, *p* = 0.967) and the post-CA ONSD (5.50 vs. 5.72 mm, *p* = 0.075). However, the amount of ONSD change in the GNO group was significantly lower than that in the PNO group (0.30 vs. 0.63 mm, *p* = 0.030). Additionally, the rate of ONSD change in the GNO group was significantly lower than that in the PNO group (5.26 vs. 12.29%, *p* = 0.041). Multivariable analysis revealed no independent association between the rate of ONSD change and poor neurologic outcome (OR = 1.075; 95% CI = 0.990–1.167; *p =* 0.084) (Table 3).

### 3.4. Diagnostic Value of ONSD Changes for Predicting the Neurologic Outcome

The AUC for predicting PNO was 0.64 (95% CI = 0.53–0.73; *p* = 0.04) in the ROC curve for the rate of ONSD change (Figure 3). Patients with a rate of ONSD change >27.2% had PNO with a specificity and PPV of 100%. GNO could be predicted using a cut-off value of ≤5.83% in the ROC curve for the rate of ONSD change, with a sensitivity and specificity of 60.0 and 76.06%, respectively; the PPV and NPV were 46.9 and 84.4%, respectively (Table 4).

## 4. Discussion

This study revealed that the amount and rate of ONSD change were significantly associated with neurologic outcomes. However, there was no significant between-group difference in the post-CA ONSD and no independent association of the rate of ONSD change with neurologic outcome after adjusting for confounding variables. Together with other established predictors, the rate of ONSD change may be useful for predicting neurologic outcomes. To the best of our knowledge, this is the first study to investigate individual differences in ONSD changes among post-CA survivors.

Previous studies have reported an association of neurologic outcomes among critically ill patients, including post-CA survivors, with increased ICP values [3,4,21,22]. The optic nerve sheath enclosing the optic nerve comprises a subarachnoid space layer, which is filled with cerebrospinal fluid (CSF) [23]; hence, ICP is positively correlated with the CSF pressure and the ONSD [23,24]. The ONSD is a potential non-invasive ICP estimator and could be useful for assessing intracranial hypertension [24]. In patients with post-CA hypoxic cerebral injury, increased ICP is associated with the neurologic outcome [3,21,22].

Several studies have reported that the ONSD can predict neurologic outcomes in post-CA survivors. A retrospective cohort study from Korea reported an association between higher ONSD values on initial brain CT and poor neurologic outcomes [14]. Chelly et al. demonstrated the potential role of the ONSD as an early prediction tool for outcomes in post-CA patients treated with TTM administration [18]. Other studies have applied the ONSD in combination with other predictors, including GWR and albumin levels, to enhance the predictive value [13,19,21]. Moreover, two recent meta-analyses reported the potential use of the ONSD in predicting neurologic outcomes [19,20]. A registry-based multicenter study demonstrated inconsistency with these previous findings, as it reported no correlation between the ONSD on early unenhanced brain CT and neurologic outcomes in post-CA survivors managed with TTM administration [25]. Furthermore, previous studies indicated a limited and insufficient role of the post-CA ONSD alone in predicting neurologic outcomes in post-CA survivors [26].

The ONSD can provide non-invasive ICP measurement and could serve as a surrogate marker for increased ICP [16,17]. However, in healthy adults, there are differences in the baseline ONSD according to individual characteristics, including sex, body mass index (BMI), race, and eyeball size [27,28]. Most studies performed on healthy volunteers have reported that the mean ONSD ranges from approximately 3 to 5 mm; furthermore, the reported mean or median ONSD values have varied across study cohorts, depending on race or measurement tools [27,28,29,30]. Ultrasonographic evaluation of healthy Asians revealed a higher ONSD value in males and individuals with a high BMI [27,28]. Therefore, these individual differences could confound the interpretation of the post-CA ONSD; furthermore, considering the baseline, ONSDs may help in improving the prognostic value. Thus, ONSD changes are potentially useful markers for ICP measurement changes. A prospective observational study on ONSD changes in patients with hydrocephalus reported a significant reduction in the ONSD after a ventriculoperitoneal shunt operation [31]. In our study, ONSD changes were more reliable than the ONSD itself in predicting neurologic outcomes in patients with post-CA.

A recent meta-analysis reported that, in comparison to CT and magnetic resonance imaging (MRI), sonographic measurement more accurately predicted neurologic outcomes among patients with post-CA [19]. However, obtaining and comparing pre-CA and post-CA ONSDs using ultrasound has limitations in clinical settings. Moreover, determining the pre-CA ONSD using brain MRI also has limitations, given its specific modality. Recent studies indicate that the axial proton density/T2-weighted turbo spin-echo fat-suppressed sequence is required for ONSD measurements using MRI. However, in most post-CA patients, the turbo spin echo image is not included in the diffusion-weighted MRI [32,33]. Additionally, there is a strong association between the ONSD and eyeball transverse diameter (ETD) (ONSD/ETD ratio) in healthy adults [34]. However, there is a need for further studies on the association between the ONSD/ETD ratio and neurologic outcomes among post-CA patients.

This study has several limitations. First, this was a single-center study with a limited sample size that led to insufficient statistical power; however, we performed a power analysis to calculate the sample size, which was relatively large compared with those of other studies. Second, this retrospective study included patients who underwent both pre- and post-CA brain CT, which could lead to selection bias affecting the results. Third, although we attempted to extensively collect variables based on the Utstein Resuscitation Registry Templates, there may still be hidden confounders [35]. Fourth, there could have been minor measurement errors given the very small size of the ONSD in brain CT. However, to minimize these errors, two blinded emergency physicians performed measurements using a standardized method showing consensus. Fifth, current guidelines recommend a neurologic outcome assessment at 3 months after discharge [10]. However, we measured the neurologic outcomes at discharge and did not determine the long-term outcomes. Sixth, OHCA and in-hospital cardiac arrest (IHCA) were both included and analyzed in this study, despite the differences in the characteristics and proportion of GNO and PNO. Follow-up studies that include only OHCA or IHCA patients may be required. This was a retrospective study, and the clinical utility of the predictive value for prognosis remains unclear. Hence, there is a need for large-scale prospective studies to confirm our findings.

## 5. Conclusions

The rate and amount of ONSD changes in brain CT were significantly associated with neurologic outcomes in patients with post-CA. ONSD changes may be useful to predict neurologic outcomes in patients with post-CA.

## Figures and Tables

**Figure 1 ijerph-18-06567-f001:**
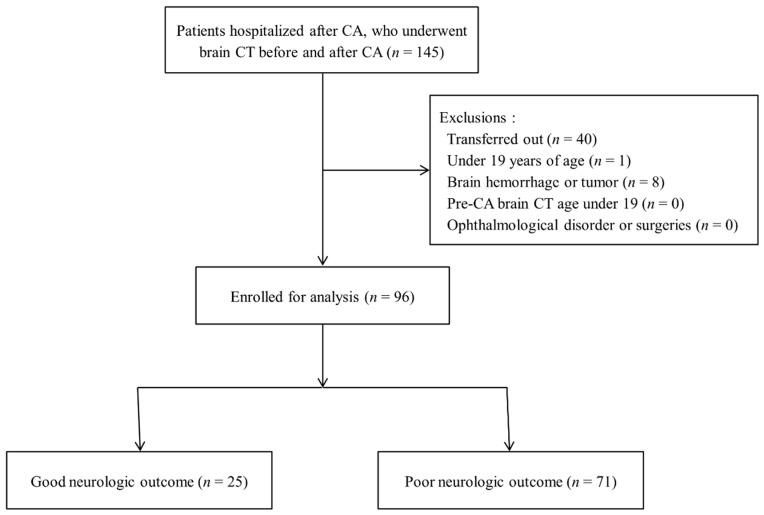
Flow chart of the study process.

**Figure 2 ijerph-18-06567-f002:**
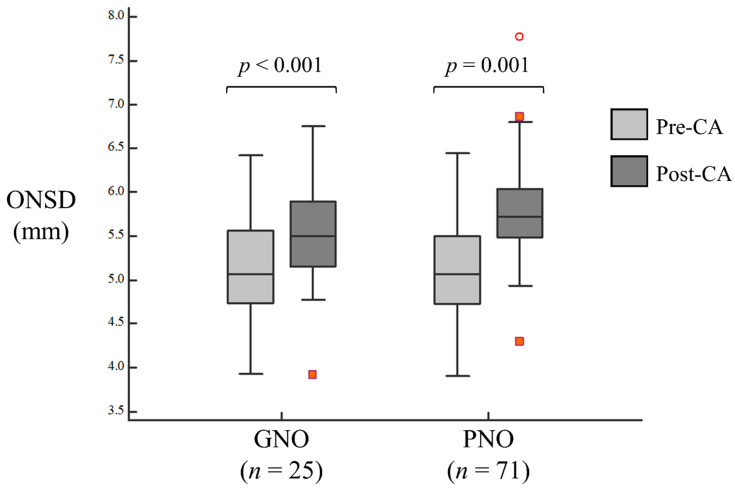
Comparison of the optic nerve sheath diameter between pre-cardiac arrest and post-cardiac arrest in good and poor neurologic outcome groups. The red circle and cubes mean outliers.

**Figure 3 ijerph-18-06567-f003:**
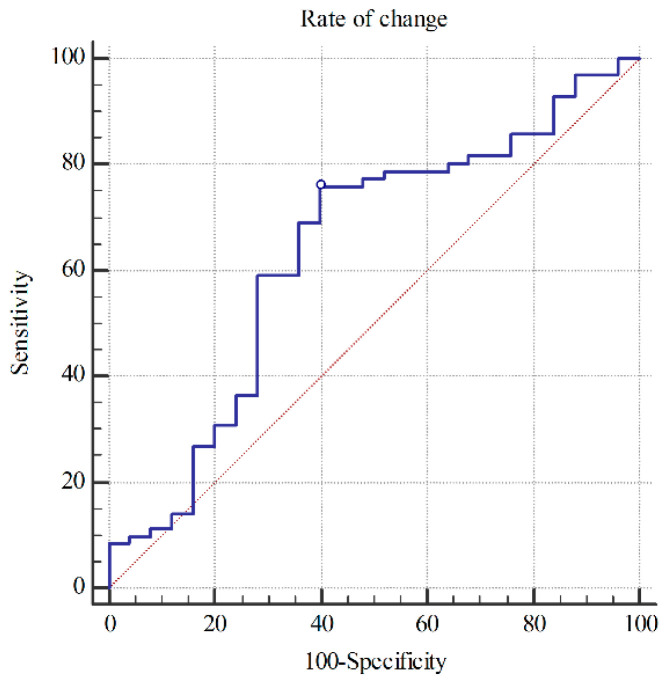
Receiver operator curve for predicting poor neurologic outcome using rate of optic nerve sheath diameter change. AUC = 0.64 (95% confidence interval = 0.53–0.73).

**Table 1 ijerph-18-06567-t001:** Baseline characteristics of enrolled patients.

	Total (*n* = 96)	GNO (*n* = 25)	PNO (*n* = 71)	*p*-Value
Demographics				
Age, year	70 (58–79)	60 (52–67)	75 (61–80)	<0.001
Sex, male	54 (56.3)	15 (60.0)	39 (54.9)	0.660
Comorbidities				
HTN	52 (54.2)	14 (56.0)	38 (53.5)	0.831
DM	37 (38.5)	6 (24.0)	31 (43.7)	0.082
MI	16 (16.7)	4 (16.0)	12 (16.9)	1.000
Etiology				
Cardiac	23 (24.0)	14 (56.0)	9 (12.7)	<0.001
Respiratory	40 (41.7)	8 (32.0)	32 (45.1)	0.254
Others	33 (34.4)	3 (12.0)	30 (42.3)	0.006
Resuscitation				
Location of arrest, OHCA	76 (79.2)	16 (64.0)	60 (84.5)	0.030
Witnessed	72 (75.0)	19 (76.0)	53 (74.6)	0.893
Bystander CPR	61 (63.5)	18 (72.0)	43 (60.6)	0.307
Shockable rhythm	11 (11.5)	8 (32.0)	3 (4.2)	0.001
No-flow time, min	10 (0–21)	4 (0–9)	11 (2–25)	0.003
Low-flow time, min	10 (6–16)	6 (3–10)	11 (8–18)	0.004
TTM	6 (6.3)	3 (12.0)	3 (4.2)	0.180
CT to ROSC interval *, month	27 (6–55)	40 (6–55)	23 (6–53)	0.780
ROSC to CT interval ^†^, min	104 (51–171)	60 (33–118)	113 (60–200)	0.017

Abbreviations: GNO = good neurologic outcome; PNO = poor neurologic outcome; HTN = hypertension; DM = diabetes mellitus; MI = myocardial infarction; OHCA = out-of-hospital cardiac arrest; CPR = cardiopulmonary resuscitation; TTM = targeted temperature management; CT = computed tomography; ROSC = return of spontaneous circulation. * The interval between the latest pre-CA brain CT and ROSC. ^†^ The interval between ROSC and post-CA brain CT.

**Table 2 ijerph-18-06567-t002:** The comparisons of the amount and rate of ONSD changes between good and poor neurologic outcomes.

	Total (*n* = 96)	GNO (*n* = 25)	PNO (*n* = 71)	*p*-Value
Optic nerve sheath diameter				
Pre-CA, mm	5.07 (4.73–5.52)	5.06 (4.76–5.53)	5.07 (4.73–5.52)	0.967
Post-CA, mm	5.66 (5.41–6.01)	5.50 (5.16–5.88)	5.72 (5.49–6.04)	0.075
Optic nerve sheath diameter changes between pre-CA and post-CA				
Amount of change, mm	0.57 (0.25–0.84)	0.30 (0.18–0.65)	0.63 (0.32–0.87)	0.030
Rate of change, %	11.10 (4.70–17.21)	5.26 (3.85–14.15)	12.29 (5.83–18.74)	0.041

Abbreviations: GNO = good neurologic outcome; PNO = poor neurologic outcome; CA = cardiac arrest.

**Table 3 ijerph-18-06567-t003:** Multivariable logistic regression analysis for poor neurologic outcome with baseline variables and rate of optic nerve sheath diameter change.

Variables	Adjusted OR (95% CI)	*p*-Value
Age, year	1.115 (1.031–1.206)	0.006
DM	3.358 (0.636–17.733)	0.154
Shockable rhythm	0.084 (0.008–0.911)	0.042
No-flow time, min	1.113 (1.003–1.235)	0.043
Low-flow time, min	1.123 (1.024–1.231)	0.013
TTM	0.119 (0.008–1.794)	0.124
Location of arrest, OHCA	0.833 (0.115–6.014)	0.856
ROSC to CT interval *, min	0.999 (0.999–1.000)	0.086
Etiology, cardiac	0.080 (0.012–0.558)	0.011
Rate of change, %	1.075 (0.990–1.167)	0.084

Abbreviations: OR = odds ratio; DM = diabetes mellitus; TTM = targeted temperature management; OHCA = out-of-hospital cardiac arrest; ROSC = return of spontaneous circulation; CT = computed tomography. * The interval between ROSC and post-CA brain CT.

**Table 4 ijerph-18-06567-t004:** Cut-off and diagnostic value of optic nerve sheath diameter change for predicting good and poor neurologic outcomes.

	Cut-Off, %	Sensitivity	Specificity	PPV	NPV
Rate of change for predicting PNO	>27.2	0.085	1.000	1.000	0.278
Rate of change for predicting GNO	≤5.83	0.600	0.761	0.469	0.844

Abbreviations: PPV = positive predictive value; NPV = negative predictive value; PNO = poor neurologic outcome; GNO = good neurologic outcome.

## Data Availability

The datasets used and analyzed during the current study are available from the corresponding author on reasonable request.

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
