# Peer review of "Predictive Utility of Changes in Optic Nerve Sheath Diameter after Cardiac Arrest for Neurologic Outcomes"

_ijerph, 2021, doi:10.3390/ijerph18126567_

Round 1
Reviewer 1 Report
No scientific interest, its only an observational study that illustrates a neurological damage without any improvement for therapy or clinical assessment. I thinks the paper shouldn’t be accept
Author Response
Dear,
Thank you for your comments. We have addressed all of your comments as follows.
Best regards.
Answers for Specific points
Thank you for reviewing our paper, and we respect your opinion.
However, clinicians are eager for additional prognostic factors that can be considered when making treatment decisions because the prognostication of neurological outcome in cardiac arrest is notoriously difficult.
The novel finding of this study, compared with that of previous studies, is that pre-cardiac arrest optic nerve sheath diameter (ONSD) was included in the analyses. A limitation of previous studies, including two recent meta-analyses, is that they do not consider changes in ONSD. Therefore, ONSD itself cannot be used as a marker to predict the prognosis of cardiac-arrest patients, and new markers such as ONSD/ETD ratio have been studied recently as indirect indicators to estimate intracranial pressure.
In our study, we measured both pre- and post-cardiac arrest ONSD and analyzed the amount and rate of changes to improve predictive accuracy of neurologic outcome. We feel that our study demonstrates that the use of ONSD changes is advantageous in predicting neurologic outcome in post-cardiac arrest patients.
Despite the relatively small sample size and limitations of our study, we believe our findings have important implications, but a further study is warranted. We thank you again and respectfully ask for further consideration.
Additionally, we provide the editing certificate.
Reviewer 2 Report
The present work shows a novel topic of great interest for the field of cardiopulmonary resuscitation.
The introduction provides background and references that put the reader in context and encourage their reading.
The research design is consistent and appropriate to the type of study.
The material and methods are adequately described, as well as the results are clearly presented.
The discussion addresses the main strengths and weaknesses of the study.
The conclusions are limited to the results obtained.
Congratulations on your work
Author Response
Thank you for your comments.
We sincerely appreciate your review of our paper and your positive feedback.
Reviewer 3 Report
- It is curious that the GNO group still being younger than the PNO group, had cardiac health problems. How do the authors explain this observation? Furthermore, in lines 143-145, the authors stated that “the GNO group showed a significantly higher frequency of cardiac etiology and shockable rhythm no-flow and low-flow time than the PNO group” but in table 3, the variables that impact the PNO group were shockable rhythm, no flow time and low-flow time. This statistic profile is similar in the GNO group?
- Considering the PNO group and GNO group, the ratio of OHCA events is 1.56 and 1.18, respectively. This ratio is very close between both groups, and probably not significant if the PNO group is increased. The authors should explore this possibility.
- Which could the role of diabetes or hypertension on modulation of the optic nerve sheath diameter?
- Similar findings have been published in different sources (https://doi.org/10.1155/2020/5219367), which is the most relevant contribution of this work compared with published papers?
Author Response
Dear,
Thank you for your detailed and rigorous comments. We considered again for all of your comments as follows. Sincerely.
Best regard.

Reviewer 4 Report
Dear Dr. Heekyung Lee, dear co-authors,
Thank you very much for your contribution to the body of knowledge around prediction of neurological outcome after cardiac arrest.
I applaud your inventive study design, where you have used imaging data from patients before and after their cardiac arrest. Sadly, you've found that, in this cohort, the ONSD parameter you studied had limited additional predictive value in a multivariate model. This by no means indicates that it is not important to study this phenomenon, and I think some of the statistical findings you did weren't significant due to small sample size.
- prognostication of neurological outcome after cardiac arrest is notoriously difficult. Clinicians are eager for additional prognostic factors that can be considered when making treatment decisions. - the authors explore the value of a measurable, simple prognostic variable: optic nerve sheath diameter. It is known to be a measure of intracranial pressure and cerebral edema, and could thus be helpful in detecting damage to the central nervous system. - the study design they used is interesting, because they were able to use pre-cardiac arrest imaging data. In clinical practice, one hardly ever has the luxury of having a pre-cardiac arrest head CT to compare with a post-cardiac arrest CT. The clinical usefulness of the parameter “ ONSD change from pre- to post-cardiac arrest” is therefore by definition very limited. However, this study is still relevant, because it can be considered a “proof of principle” study. In fact, the authors show that the ONSD change is probably correlated to duration of no/low flow time (and thus how long the brain has been exposed to hypoxia). - the observation that very large changes in ONSD are have a good PPV for poor neurological outcome is relevant, and warrants further research. I can imagine this finding to be relevant for low-resource settings, where ONSD could be measured or monitored over time using ultrasound, as described in the brain trauma literature. I have not found any serious methodological flaws that warrant major revisions. Also, the authors have shown enough understanding of the limitations of their findings. I think the paper is acceptable for publication in its current form.
I wish you good luck in your scientific career.
Author Response
Thank you for your comments.
We sincerely appreciate your review of our paper and your positive feedback.
Additionally, we provide the editing certificate.

Round 2
Reviewer 1 Report
little impact on clinical practice. statistical analysis was well done but complex data required.
Reviewer 3 Report
No more comments